chemical physics/materials science

ethanolamine, Bi$_2$WO$_6$, microwave, photocatalytic activity, morphology

**Author for correspondence:**
Xingchen Liu
e-mail: liustxc@163.com

This article has been edited by the Royal Society of Chemistry, including the commissioning, peer review process and editorial aspects up to the point of acceptance.

# The three-dimensional flower-like Bi$_2$WO$_6$ assisted by ethanolamine through a microwave method for efficient photocatalytic activity

## Xingchen Liu, SuZhen Wang, Song Wang, Han Shi, Xiaolong Zhang and Zhicheng Zhong

Hubei Key Laboratory of Low Dimensional Optoelectronic Materials and Devices, School of Physics And Electronic Engineering, Hubei University of Arts and Science, Xiangyang, 441053, Hubei, People's Republic of China

XL, 0000-0002-8516-2759

The three-dimensional flower-like Bi$_2$WO$_6$ was synthesized through a one-step microwave method (the reaction temperature was 434 K and the reaction took 10 min) with the assistance of ethanolamine (EA). The prepared samples were characterized by X-ray diffraction, scanning electron microscopy, Fourier-transform infrared spectroscopy, ultraviolet–visible spectroscopy, PL, X-ray photoelectron spectroscopy and Brunauer–Emmett–Teller analysis. Methyl orange was used as target pollutant to evaluate the photocatalysis property of samples. Furthermore, the influence of the mechanism of EA on the structure and catalytic performance of Bi$_2$WO$_6$ was discussed. The detailed characterizations revealed that the three-dimensional flower-like Bi$_2$WO$_6$ was successfully synthesized with the assistance of EA. The results confirmed that EA significantly influenced the morphology of Bi$_2$WO$_6$ products. The addition of EA can effectively alter the pressure of the reaction and improve the crystal phase and structure of Bi$_2$WO$_6$ photocatalysts, enhancing the photocatalytic activity of samples and improving the photocatalytic efficiency. EA can serve as an assembling agent and structure-directing agent resulting in the formation of flower-like architectures. With the increase of the amount of EA, the as-prepared Bi$_2$WO$_6$ sample gradually forms a flower-like structure, leading to a shorter time of light holes migrating to the surface of the catalyst. It makes the compound rate significantly decreased, and improves the photocatalytic efficiency of the sample.

# 1. Introduction

Recently, organic pollution has become a subject of extensive scientific investigation. Since Fujishima first reported titanium dioxide as a photocatalytic for $H_2$ evolution from water [1], using semiconductors to degrade organic wastewater has become a feasible way to solve such water pollution. Of many semiconductor catalysts, $Bi_2WO_6$ has attracted the attention of researchers, owing to its unique advantage and wide application in the field of photocatalysis [2–5]. As a typical bismuth semiconductor, the structure of $Bi_2WO_6$ is easy to control [6].

It is well known that the photocatalytic properties of semiconductor catalyst materials have an important relationship with their crystal shape, morphology and particle size [7–9]. The different preparation methods of $Bi_2WO_6$ and the different additives in the synthesis process make the morphology and size of samples different, which will greatly affect the photocatalytic activity of samples. In traditional hydrothermal synthesis, it often takes a longer time [10,11]. However, microwave technology has the characteristics of fast, high efficiency and homogenization [12–16]. Since the heating rate of the microwave method is fast, which avoids the abnormal growth of crystal particles in the process of material synthesis, the materials with high purity, fine particle size and uniform distribution can be prepared in a short time and low temperature [17,18]. Many relevant studies have shown that organic additives can effectively control the shape and performance of the samples through selective adsorption and subsequent removal process [3,19]. Qi *et al.* used double-hydrophilic block copolymers as crystal growth modifiers to direct the controlled precipitation of barite [20]. Hu *et al.* [21] successfully synthesized nickel sulfide (NiS) hollow spheres through polymethyl methacrylate. However, the organic additives used in many synthetic studies are expensive long chain molecules. As a monoamine with one N-chelating atom, ethanolamine (EA) can be also used as a structure-directing agent for the growth of metal chalcongenide one-dimensional nanostructures under mild solvothermal reaction conditions [22]. As far as we know, it is rare to use EA as a surface modifier in $Bi_2WO_6$ semiconductor photocatalyst.

In this paper, we reported a facile and tunable synthesis of $Bi_2WO_6$, with different morphologies via an EA-assisted microwave process. Herein, EA was introduced as an assembling and structure-directing agent to controllable synthesis of $Bi_2WO_6$ architectures without any other long-chain organic molecules assisting in a one-step microwave process. The whole synthesis process took only 10 min. The morphology and photocatalytic properties of the synthetic samples were investigated, and the action mechanism of EA was discussed.

# 2. Experimental

## 2.1. Synthesis of photocatalysts

The photocatalysts were prepared in a one-step microwave process as follows: 2 mmol of bismuth nitrate and 1 mmol of sodium tungstate were added to a mixture solution with a certain proportion of EA and water, stirring evenly. The total volume of EA (99%) and water remained 6 ml. The mixed solution was then transferred to a microwave reactor (Biotage Sweden) at 434 K, and the synthetic reaction was performed for 10 min. The resulting samples were received after centrifugal washing and baking at 354 K for 10 h in the oven. The as-prepared samples are recorded as EA:X, where X is the volume ratio of EA and water.

## 2.2. Characterization

The X-ray diffraction (XRD) patterns were obtained on a D8 Advance X-ray diffractometer equipped with Cu-K$\alpha$ radiation. The morphology of the samples was explored using a Hitachi S4800 scanning electron microscope (SEM). Fourier-transform infrared spectroscopy (FTIR) was recorded from KBr pellets in the range of $400-4000 \, cm^{-1}$ on a Nicolet iS50 FTIR spectrometer. PL spectra were accomplished in solids with a Fls-980 Edinburgh fluorescence spectrophotometer with an excitation wavelength of 380 nm. X-ray photoelectron spectra (XPS) were acquired on an American electronics physical HI5700ESCA system with Al K$\alpha$ (1486.6 eV) as the excitation source.

## 2.3. Photocatalytic test

Methyl orange (MO) is stable and does not decompose under the action of light. Moreover, the degradation rate of MO can be calculated by absorbance and concentration standard curves. Therefore, MO is used us a

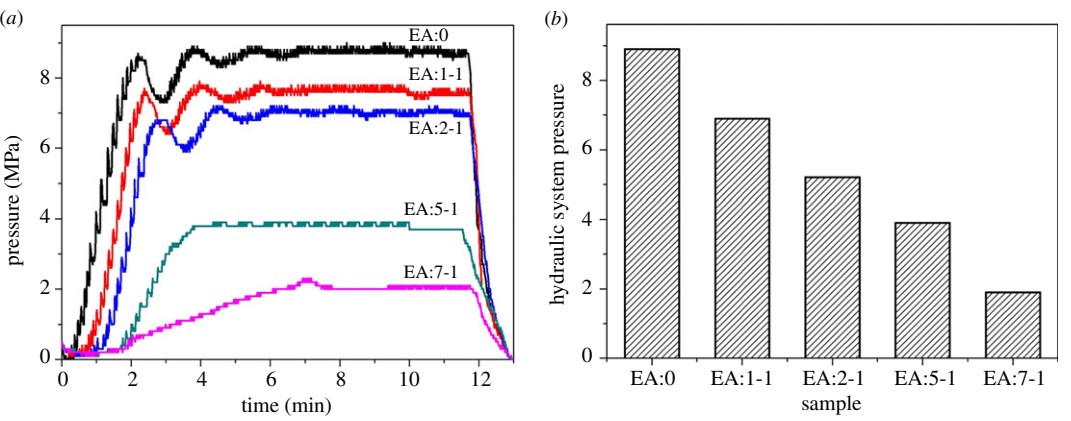

**Figure 1.** (*a*) The pressure diagram and (*b*) the maximum pressure diagram of the reaction system during the synthesis process.

model for photocatalytic experiments. The photocatalytic activity of the prepared materials was investigated by the reduction of MO in a 300 W xenon lamp with main emission wavelength 365 nm (the light intensity was 250 mW cm$^{-2}$). The distance between light source and solution surface was 18 cm. This catalytic experiment was carried out on a photocatalytic reactor. Ten-milligram samples were added to a solution of 100 ml MO (20 mg l$^{-1}$) (pH = 7).The adsorption–desorption equilibrium of catalyst and pollutant were achieved by stirring in the dark environment for 30 min. During illumination, about 5 ml of suspension was taken from the reactor at a scheduled interval. The concentration of MO was determined by absorbance analysis at 464 nm.

# 3. Results and discussion

## 3.1. Pressure of the reaction

Figure 1 shows the diagram of the system pressure detected by the automatic microwave reaction system in the process of synthesizing samples. As seen from the figure, the quantity of EA added to the solution has a significant effect on the stress of the system (figure 1*b*). As the amount of EA increases, the pressure of the reaction system becomes smaller. The boiling point of EA is about 444 K [23], which is higher than that of water (374 K). The temperature used in the synthesis process is 434 K. The proportion of EA increases, leading to increasing of the boiling point of the reaction mixture, therefore the reaction pressure decreases. As we all know, the effect of pressure on the reaction speed is obvious during chemical reaction [24,25]. EA can adjust the system pressure, resulting in the changed reaction speed. In addition, the viscosity of EA is much higher than that of water, which also affects the transmission rate of ions in the system [26], and thus effectively regulates the reaction speed. In general, EA can effectively control the reaction rate in the synthesis of Bi$_2$WO$_6$.

## 3.2. XRD patterns

Figure 2 shows the XRD pattern of EA:X samples. The sample without EA only has a broad and weak peak centred at $2\theta = 28°$, which shows that when there is only water in the reaction solution, the crystals of the resulting sample are not good. When the ratio of EA and water is 1 : 1, the diffraction peaks at 27.88°, 32.36°, 46.40°, 54.95°, 57.60° and 67.58° can be attributed to Bi$_2$O$_3$, according to PDF card JCPDS no.74−1373 (asterisk '*'), and those at 28.08°, 33.03°, 47.21°, 56.06°, 58.59°, 69.04° are due to the crystalline Bi$_2$WO$_6$ (JCPDS no. 39-0256). The result shows that EA:1-1 is the mixture of Bi$_2$O$_3$ and Bi$_2$WO$_6$. In addition, when only Bi precursor is used in a similar experimental condition, Bi$_2$O$_3$ is obtained (JCPDS no.65-2366; electronic supplementary material, figure S1). As for EA:2-1, EA:5-1 and EA:7-1, no diffraction peaks characteristic of Bi$_2$O$_3$ are observed, revealing that EA:2-1, EA:5-1 and EA:7-1 are all pure Bi$_2$WO$_6$. When the reaction time is prolonged for EA:1-1, the diffraction peaks characteristic of Bi$_2$O$_3$ gradually decreased, and that of Bi$_2$WO$_6$ enhanced gradually (electronic supplementary material, figure S2). As shown in figure 1, the addition of EA can decrease the rate of reaction. Moreover, the high dispersion of EA makes the precursor sodium tungstate and bismuth

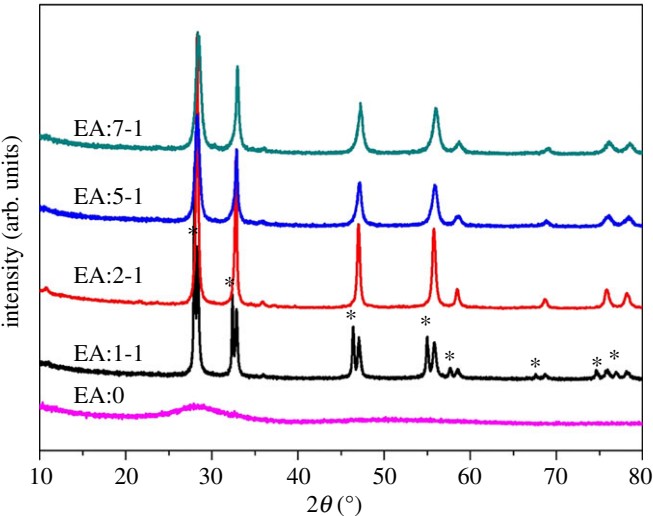

**Figure 2.** XRD pattern of EA:X samples.

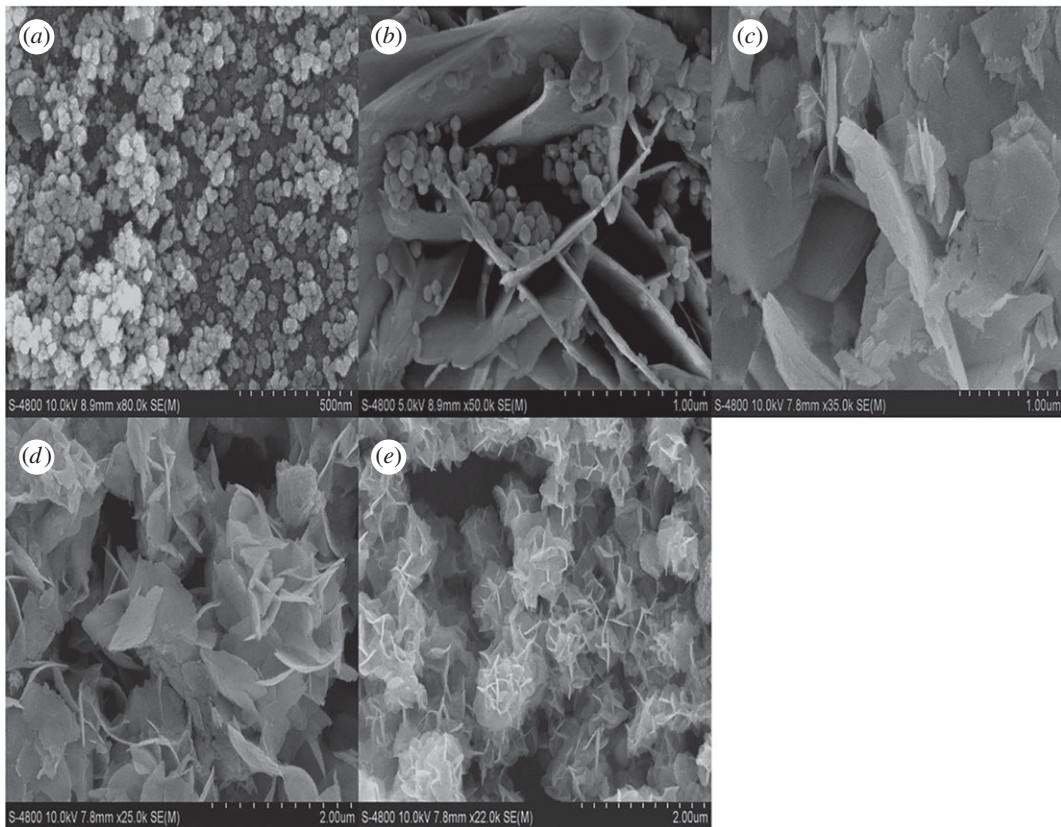

**Figure 3.** SEM images of EA:X samples: (*a*) H$_2$O, (*b*) EA:1-1, (*c*) EA:2-1, (*d*) EA:5-1 and (*e*) EA:7-1.

nitrate fully in contact, thus contributing to the formation of Bi$_2$WO$_6$ [27]. The results show that the addition of EA is beneficial to the formation and growth of Bi$_2$WO$_6$ crystalline phase.

## 3.3. SEM patterns

Figure 3 shows the SEM patterns of EA:X samples. When no EA is added, the sample presents a small granular structure. The particle size is not uniform, and the average particle size is about 10 nm (figure 3*a*). The irregular flaky structure is beginning to appear in EA:2-1 (figure 3*b*), the structure of which is the mixture of polyhedron and irregular flakes. This result proves that EA:1-1 is a mixture of

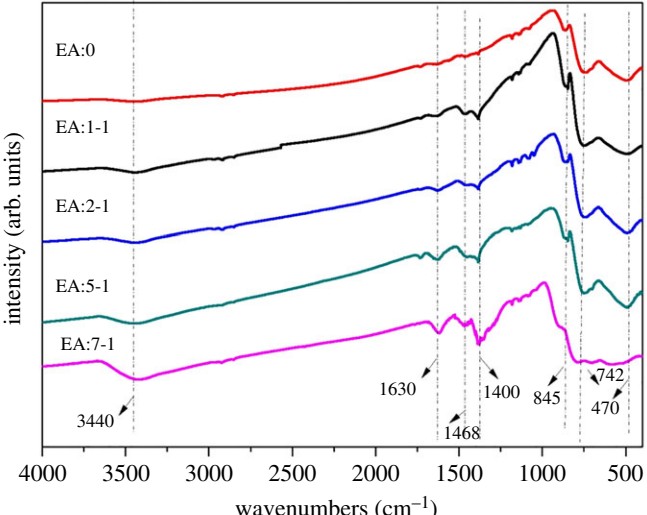

**Figure 4.** FTIR images of EA:X samples.

two substances, which is consistent with the XRD result. When the proportion of EA and water reaches 2:1 (the sample is pure $Bi_2WO_6$), the polyhedral structure disappears, and irregular thin sheets form (figure 3c). When the addition of EA is further increased, the two-dimensional flakes gradually accumulate into the three-dimensional flower ball structure, and the size of the sample decreases (figure 3d). As the EA and water ratio is as high as 7 : 1, the flower ball structures of $Bi_2WO_6$ are more obvious. EA:7-1 is of uniform size, and highly fragmented (figure 3e). The three-dimensional flower-like structure of $Bi_2WO_6$ can reduce the time required for the migration of photo-holes to the surface of the catalyst, resulting in a significant reduction in the recombination rate of electron and hole, thus improving the photocatalytic efficiency of the samples [28]. The results show that EA can regulate the formation and growth of the synthesized $Bi_2WO_6$ crystal nucleus, thus controlling the morphology of $Bi_2WO_6$.

## 3.4. FTIR patterns

The FTIR spectra of the EA:X samples synthesized are shown in figure 4. The band at 1630 cm$^{-1}$ is the characteristic peak of free water. The bands at both 845 and 470 cm$^{-1}$ belong to Bi-O vibration [29] and the peak at 722 cm$^{-1}$ is related to W-O vibration [28]. The broad peak ranging from 3200 to 3550 cm$^{-1}$ may be related to −OH cluster peaks stretching, with the proportion of EA increasing, which becomes wider gradually, and moves to a lower wavenumber. The results show that the association of −OH and $Bi_2WO_6$ molecularly becomes greater with the proportion of EA increasing through hydrogen bonding. In addition, the peak at 1400 cm$^{-1}$ is related to the bending vibration peak of −NH$_2$. And the peak intensity of this peak increases with the proportion of EA. One possible reason is that the EA molecule contains an active primary hydroxyl group and secondary amino group. In the process of the synthesis of $Bi_2WO_6$, −OH and −NH$_2$, the EA molecules are attached to the samples of $Bi_2WO_6$. With the addition of EA, more −OH and −NH$_2$ groups are attached to $Bi_2WO_6$, and the peak strength increases.

## 3.5. UV–Vis patterns

The UV–Vis spectra of EA:X are measured to evaluate light absorption property. As shown in figure 5a, after EA is added, a significant red-shift of the absorption edge of EA:1-1 to longer wavelength is observed. As the amount of EA increases, the sample becomes pure $Bi_2WO_6$ (EA:2-1, EA:5-1 and EA:7-1 samples), and the maximum absorption edge of the sample moves towards the visible region.

As we all know, the red-shift of the wavelength always means band gap narrowing of the photocatalysts. Therefore, the plots of $(\alpha h\nu)^2$ versus $h\nu$ are used to determine the band gaps' value of the as-prepared photocatalysts. As shown in figure 5b, the values of band gaps of EA:0, EA:1-1, EA:2-1, EA:5-1 and EA:7-1 are determined to be about 2.72 eV, 2.44 eV, 2.64 eV, 2.46 eV and 2.36 eV, respectively. EA:1-1 sample is the mixed crystal phase of $Bi_2O_3$ and $Bi_2WO_6$, furthermore the bandgap value of $Bi_2O_3$ is smaller than that of $Bi_2WO_6$ [30,31], leading to the smaller bandgap of EA:1-1,

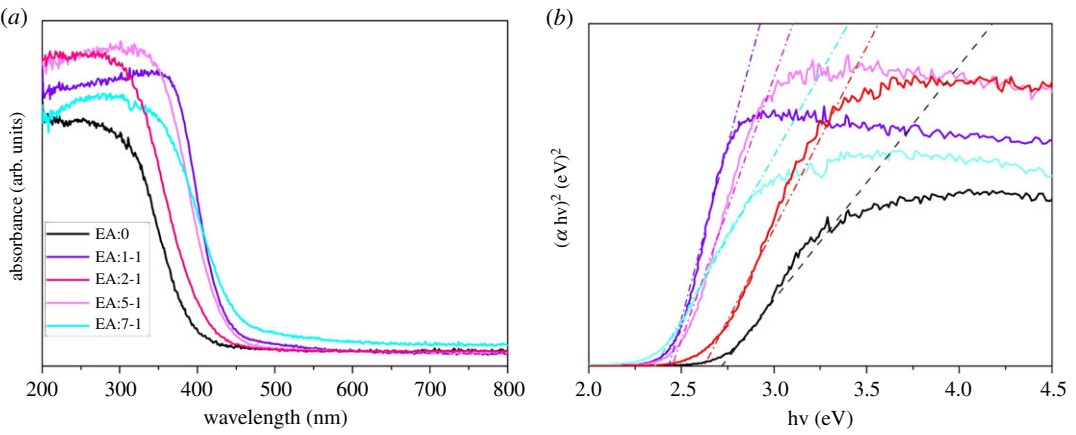

**Figure 5.** (*a*) UV–Vis and (*b*) DRS images of EA:X samples.

compared with EA:2-1. When the sample is pure $Bi_2WO_6$ (EA:2-1, EA:5-1 and EA:7-1 samples), with the increase of the amount of EA, the width of the band gap increases gradually. This may be due to the decrease of sample size and the increase of semiconductor defects with the increase of density, leading to the delocalization of molecular orbitals around the electronic band edge. Therefore, the absorption spectrum of wavelength moves towards long wave. Decreased band gap is beneficial to enhancing the photocatalytic activity of $Bi_2WO_6$.

## 3.6. XPS patterns

The surface compositions and chemical state of the atoms in the catalysts are further investigated by XPS measurements. Figure 6 gives the high-resolution XPS spectra of EA:X. The binding energies obtained in the XPS analysis are corrected for specimen charging by referencing C 1 s to 284.50 eV. The Bi 4f peaks of all EA:X samples at *ca* 164.00 eV and *ca* 158.63 eV, are corresponding to the inner electrons of Bi 4f 5/2 and Bi 4f 7/2, which indicates that all Bi elements in EA:X samples exist in the form of $Bi^{3+}$. The W 4f peaks of EA samples appear at *ca* 37.06 eV and *ca* 34.86 eV, demonstrating the valence state of W is +6 [6]. High-resolution spectra show that the O 1s peak can be divided into two peaks (figure 6*c*). The O 1s core level at 532.25 eV ($O_S$) can be ascribed to the oxygen in −OH, and the binding energy located at 529.70 eV belongs to the lattice oxygen ($O_L$) in $Bi_2WO_6$ [32]. Compared with EA:2-1, the atomic ratio of $O_S$ in EA:7-1 to the total oxygen ($O_S + O_L$) increases from 0.16% to 47.2%, indicating that the number of the absorbed −OH groups increases, which agrees with the FTIR results.

## 3.7. Photocatalytic properties

The photocatalytic performances of the as-prepared photocatalysts are tested by MO degradation under simulated sunlight irradiation, as shown in figure 7. With the increase of the amount of EA, the photocatalytic activity of the sample gradually improves. When the proportion of EA increases to 7 : 1, the concentration of MO in the reaction solution reduces to 0 within 60 min fundamentally. Moreover, there is a certain degree of dark adsorption capacity enhancement, with the proportion of EA increasing. The surface area of all the samples is similar, as shown in table 1, which can indicate that the difference in adsorption capacity is not caused by the BET difference. Figure 7*b* shows UV–Vis spectrum changes of the degradation of MO by the EA:7-1 catalyst. As time changes, the maximum absorption peak of MO is reduced at a rapid speed. After 60 min, the curve tends to a straight line, which shows that MO is completely degraded. The results show that the addition of EA can improve the photocatalytic activity of materials. With the increase of the amount of EA, the photocatalytic activity gradually increases.

It is important to investigate the active species in the photocatalytic process in order to understand the mechanism of photocatalysis. In general, many active species, including $\cdot OH$, $h^+$ and $\cdot O_2^-$, can be expected to exist in a photocatalytic process [33]. In this case, isopropanol (IPA), triethanolamine (TEOA) and $N_2$ purging are used as $\cdot OH$, $h^+$ and $\cdot O_2^-$ scavengers, respectively. As shown in electronic supplementary material, figure S3, the photocatalytic activity of EA:7-1 nanocomposite is greatly suppressed after the addition of TEOA, suggesting that $h^+$ is the main reactive species. Similarly, the

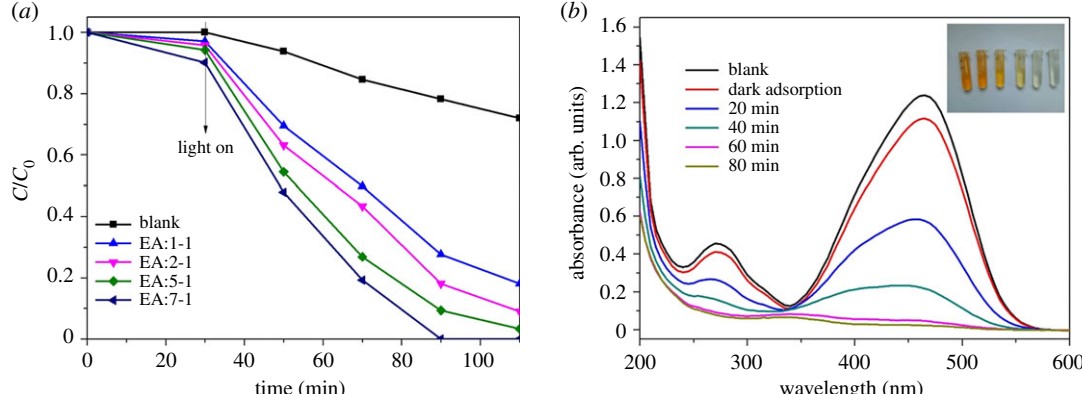

**Figure 6.** The XPS profiles of EA:X samples for (*a*) Bi 4f, (*b*) W 4f and (*c*) O 1s.

**Figure 7.** (*a*) The photocatalytic degradation curves and (*b*) the absorption spectrum of MO of the EA:7-1 sample under simulated sunlight irradiation over time.

obvious decrease in the photocatalytic activity observed by the addition of IPA and $N_2$ purging, respectively, suggest that $\cdot OH$ and $\cdot O_2^-$ play an important role in the reaction process, too.

PL spectra are derived from the composite of photogenic electrons and holes pairs, so PL analysis is usually used to examine the migration, transfer and separation of photonic carriers [34]. As shown in

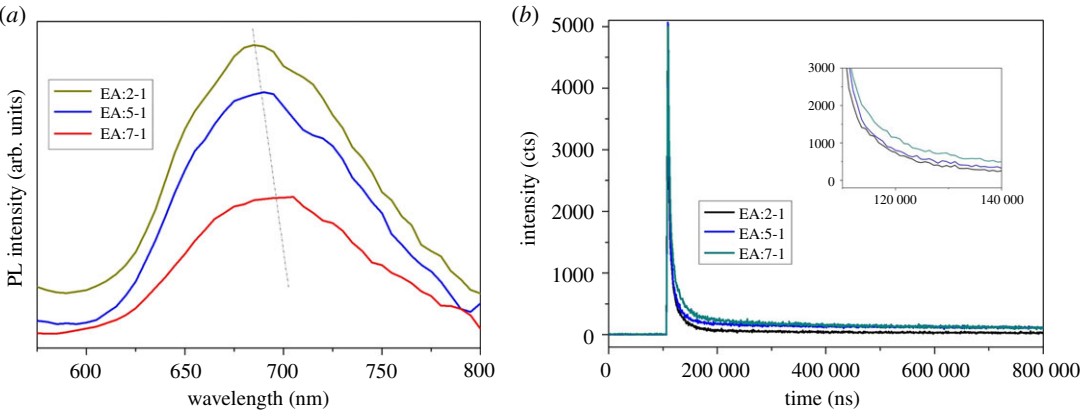

**Figure 8.** (a) PL spectra of EA:X samples and (b) fluorescence decay curves of EA:X samples.

**Table 1** BET of the EA:X samples.

| sample | $S_{BET}$ (m$^2$ g$^{-1}$) |
| --- | --- |
| EA:2-1 | 19.25 |
| EA:5-1 | 19.74 |
| EA:7-1 | 19.76 |

figure 8a, with 380 nm light excitation, a broad visible PL band, ranking from 580 nm to 800 nm, can be found for the $Bi_2WO_6$ samples. In principle, the higher the fluorescence intensity of the photoluminescence of the sample, the greater the compound probability of its photogenic electron and hole [35]. With the addition of EA increasing, the PL intensity of the sample decreases, indicating that the composite probability of the electron and hole of the $Bi_2WO_6$ sample is greatly reduced. Compared with EA:2-1, the peak of EA:7-1 is slightly red-shifted. This may be attributed to the hierarchical structure, which can act as a bridge for the electron transporter. The decay spectra of EA:X samples are shown in figure 8b. In the fluorescence attenuation test, the decrease of fluorescence intensity is due to the transition from the excited state to the ground state [36,37]. When the ratio of EA increases to 7 : 1, the decay spectra is relatively smoother, indicating that the process of photo-induced change from the excited state to ground state slows down, that is, the composite time for electrons and holes needed becomes longer. The addition of EA makes the electron-hole pairs transferring effectively achieved.

## 3.8. Possible action mechanism of EA

To confirm the band structure, valence-band XPS was carried out to analyse the valence band (VB) potential (figure 9). For EA:7-1 and EA:5-1, the VB potentials are 1.77 eV and 1.81 eV, respectively, which are smaller than those of EA:2-1 (1.97 eV). As shown in figure 10, EA:2-1, EA:5-1 and EA:7-1 exhibit the positive slope in the linear region, indicating they are n-type semiconductors. Based on these plots, the flat band potentials are estimated to be $-0.67$ V, $-0.65$ V, $-0.59$ V for EA:2-1, EA:5-1 and EA:7-1, respectively. It is clear that the electronic structure of the $Bi_2WO_6$ sample can be tailored just from EA adjustment. With the addition of EA, the VB edges increase and the CB edges decrease. The narrowed band gap of samples can be ascribed to semiconductor defects, which may extend the delocalized p-electron system.

To further confirm the optimal addition amount of EA, we increased the ratio of EA to water. As shown in the electronic supplementary material, figure S4, too much EA will lead to the decrease of photocatalytic efficiency of samples. The possible reason is that too much EA will further reduce the size of the sample, resulting in an increase in the recombination rate of electrons and holes.

In order to further study the effects of amino and hydroxyl groups in EA on the process of microwave synthesis of $Bi_2WO_6$ samples, we used ethylene glycol (EG) and ethylenediamine (ED) as reaction solutions. Figure 11 shows XRD patterns of a sample with different reaction solutions. All the reaction solutions are adjusted for pH = 10, since the pH value of the solution is 10, in which the ratio of EA and $H_2O$ is 7 : 1. As for ED:7-1, the peak can be attributed to Bi (JCPDS no.44-1426) and $Bi_2WO_6$ (JCPDS

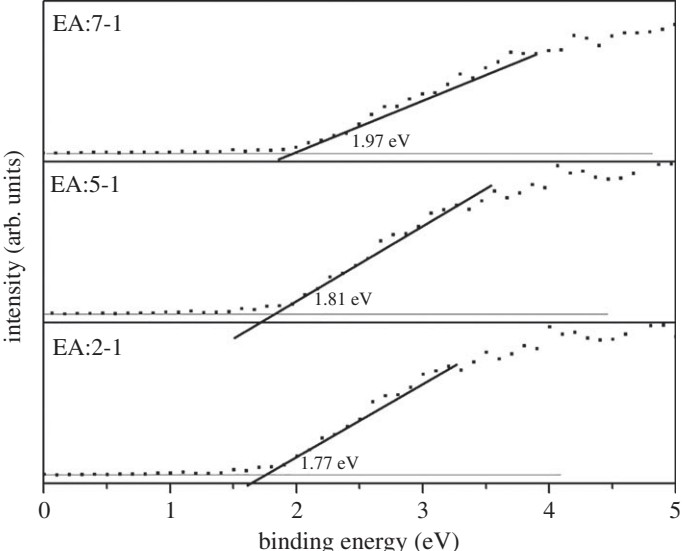

**Figure 9.** The XPS valence band spectra of EA:X samples.

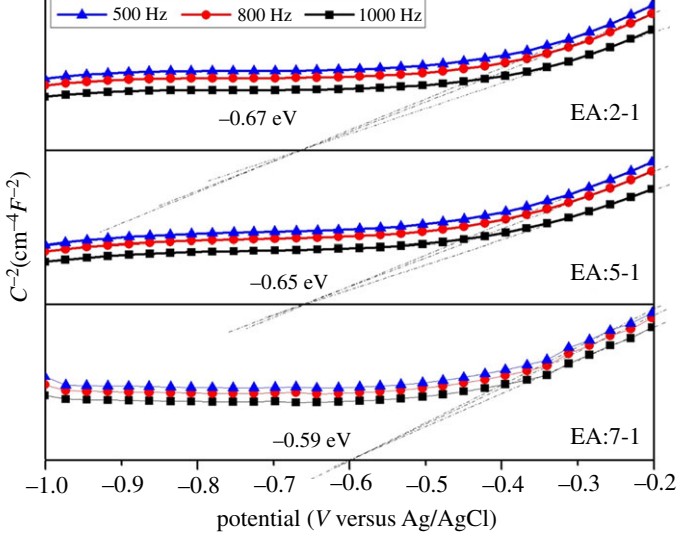

**Figure 10.** Mott–Schottky plots of EA:X.

no.39-0256), while EG:7-1 only has a broad and weak peak centered at $2\theta = 28°$. The crystal of EG:7-1 is not good, which indicates that $-NH_2$ can promote the contact of $Bi^{3+}$ and $W^{6+}$, improving the crystallinity of the sample. However, since $-NH_2$ is reducible [38], excessive $-NH_2$ makes $Bi^{3+}$ reduced to $Bi^0$.

As shown in electronic supplementary material, figure S5, the intensity of the FTIR band ranging from 3200 to 3550 cm$^{-1}$, which belongs to $-OH$ cluster peaks stretching, is in the order of EG:7-1 > EA:7-1 > ED:7-1, since that at 1400 cm$^{-1}$ (belonging to the bending vibration peak of $-NH_2$) is in the order of ED:7-1 > EA:7-1 > EG:7-1. Since there are two $-OH$ in the glycol molecule and the ethanediamine molecule contains two $-NH_2$, this leads to more $-OH$ groups absorbed to EG:7-1 and more $-NH_2$ groups absorbed to ED:7-1. Electronic supplementary material, figure S6, shows the photocatalytic degradation curves of MO of prepared samples with different additives under simulated sunlight irradiation over time. In the dark adsorption process, the adsorption capacity of ED:7-1 is much higher than that of EA:7-1, which is due to more $-OH$ and greater surface area. However, EG:7-1 has basically no photocatalytic activity.

Scheme 1 shows the schematic of the growth mechanism of samples' architecture. In the process of microwave synthesis, the high viscosity and high boiling point of EA can effectively control the reaction rate and promote the formation and growth of $Bi_2WO_6$. With the increase of the amount of EA, the as-prepared $Bi_2WO_6$ sample gradually forms a flower-like structure, leading to a shorter time for light holes migrating to the surface of the catalyst. It makes the compound rate significantly decreased, and improves

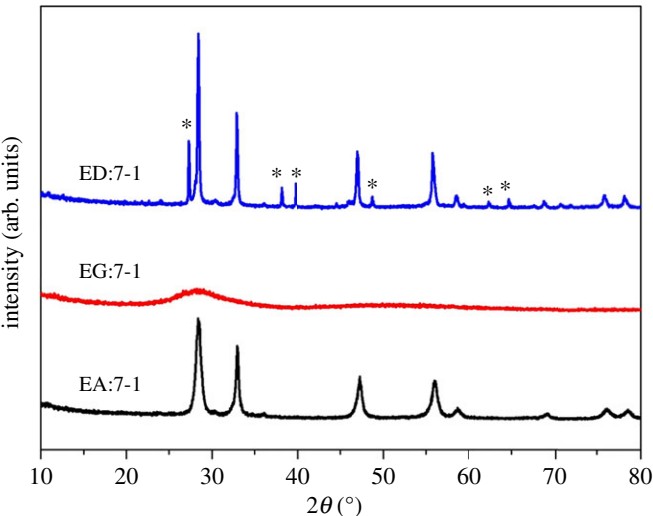

**Figure 11.** XRD patterns of prepared samples with different additives.

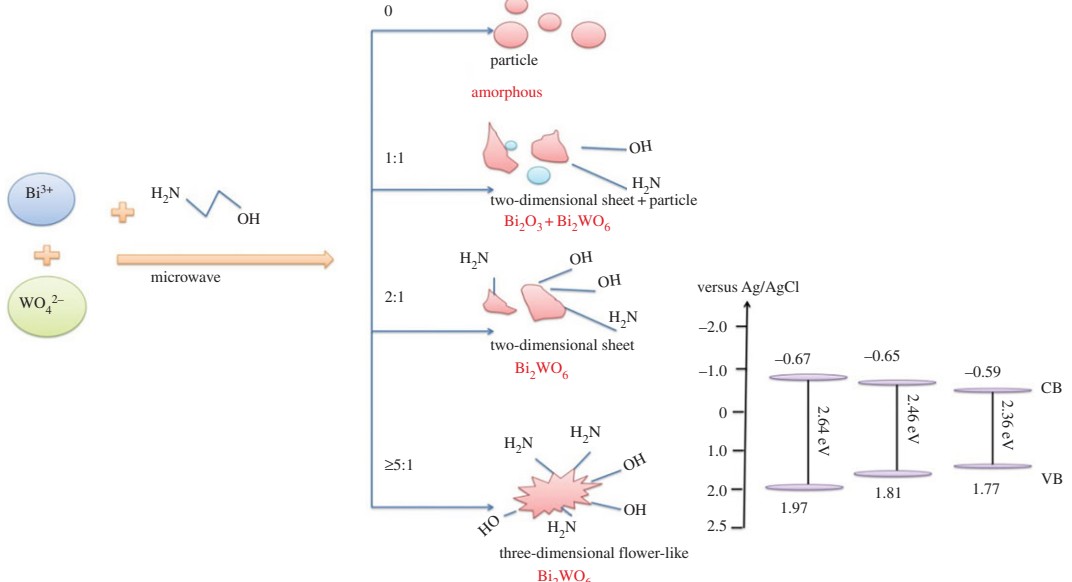

**Scheme 1.** Schematic of the growth mechanism of samples' architecture.

the photocatalytic efficiency of the sample. In addition, the size of the sample decreases, and the number of semiconductor defects increases with the increase of density, leading to the absorption spectrum of wavelength moving towards long wave. Therefore, the VB edges increase and the CB edges decrease. The narrowed band gap makes the utilization rate of light increase, leading to the enhanced photocatalytic activity. Moreover, EA molecules contain active hydroxyl and secondary amino groups; $-NH_2$ can promote the contact of $Bi^{3+}$ and $W^{6+}$, and the absorbed $-OH$ groups can serve as the adsorption site, leading to more MO molecules absorbed on the photocatalyst. In conclusion, EA can serve as an assembling agent and structure-directing agent, adsorbed on the surface of $Bi_2WO_6$ nuclei and resulting in the formation of flower-like architectures. The addition of EA can effectively improve the catalytic activity of $Bi_2WO_6$ photocatalyst and improve the photocatalytic efficiency.

# 4. Conclusion

The good performance of $Bi_2WO_6$ photocatalyst is successfully achieved by a microwave synthesis method. The method has good potential applications in the field of light catalyst synthesis, due to the fast reaction and easy operation. The high viscosity and high boiling point of EA can effectively

improve the crystal phase and structure of $Bi_2WO_6$ photocatalyst and enhance the photocatalytic activity. Moreover, EA can serve as an assembling agent and structure-directing agent, adsorbed on the surface of $Bi_2WO_6$ nuclei and resulting in the formation of flower-like architectures. These findings could be helpful in the future development of $Bi_2WO_6$ nanomaterials with different morphologies.

Ethics. We have received ethical approval from a local ethics committee to carry out this study and received informed consent for the participants to participate in the study.

Data accessibility. The data have been provided as the electronic supplementary material.

Authors' contributions. X.L. made substantial contributions to conception and design and analysis and interpretation of data; S.W. conceived the study, designed the study, coordinated the study and helped draft the manuscript; S.W., H.S. and X.Z. carried out the chemistry laboratory work and data collection. Z.Z. helped draft the manuscript. All authors gave final approval for publication.

Competing interests. We have no competing interests.

Funding. This research is financially supported by the Project of Hubei University of Arts and Science (XK2018029) and the National Natural Science Foundation of China (grant no. 21401053).

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
