## [Reviewer comments · Royal Society Open Science]

Review History

RSOS-181422.R0 (Original submission)

Review form: Reviewer 1 (Girish Kumar)

Is the manuscript scientifically sound in its present form?

Yes

Are the interpretations and conclusions justified by the results?

Yes

Is the language acceptable?

No

Is it clear how to access all supporting data?

Yes

Do you have any ethical concerns with this paper?

No

Have you any concerns about statistical analyses in this paper?

I do not feel qualified to assess the statistics

Recommendation?

Accept with minor revision (please list in comments)

Comments to the Author(s)

Dear authors,

The specific insights into the exceptional role of ethanolamine on the morphological evolution of Bi₂WO₆ sheets and further assembling into flower-like structure are explained in detail. The further discussions related to the shifting of band edge positions, photoluminescence spectroscopy and Mott-Schottky plots that substantiates the influence of ethanolamine are the other interesting aspects. However, few clarifications and thorough language editing are still essential for improved understanding.

Major comments

(1) The recycling ability of the photocatalyst may be discussed.

Minor comments

Title: It may be modified for reader's attention (up to the author's choice): (i) Pivotal role of ethanolamine on the structure-electronic and photocatalytic properties of flower-like Bi₂WO₆; (ii) Role of ethanolamine on the morphological evolution of 3-D flower like Bi₂WO₆ photocatalyst by microwave method; (iii) Formation mechanism of ethanolamine assisted 3-D flower like Bi₂WO₆ photocatalyst under microwave irradiation; (iv) 3-D flower like Bi₂WO₆: Insight into the role of ethanolamine on the formation mechanisms and photocatalytic property.

Abstract

(1) Indicate the reaction temperature and time within the parenthesis after the sentence (microwave method) and specify concentration of ethanolamine for clarity.

(2) The role of ethanolamine in altering the pressure of the reaction and on the crystallization of Bi₂WO₆ must be highlighted.

(3) The results derived from materials characterization should be briefly emphasized.

Section 1.0

(1) Modify the starting phrase 'In recently, pollution!' (avoid using pollution twice in the same phrase).

(1) Add few review articles related to Bi₂WO₆ along with ref. [2,3]: Applied Surface Science 355 (2015) 939-958; Catalysis Science and Technology 2 (2012) 694-706; Chemical Society Reviews 43 (2014) 5276-5287.

(2) Remove 'more than 10 hours' and generalize as 'longer time', while discussing hydrothermal method.

Section 2.1

(1) The preparative method should be detailed out completely to ensure reproducibility: (i) volume of Bi precursors and tungstate precursors taken should be indicated; (ii) concentration of ethanolamine must be specified.

(2) The apparatus in which these solutions were mixed must be indicated.

(3) The samples may be abbreviated as 'EA:X' instead of 'EA X'.

Section 2.3

(1) The details of excitation source including its intensity, wavelength maxima and distance

between light source and solution surface must be provided.

(2) Specify the solution pH at which degradation reaction were carried out.

Section 3.1

(1) The effect on pressure on the reaction speed as referenced from literature [15,16] may be briefly explained.

(2) The viscosity of the ethanolamine-water mixture (EA 1-1 to EA 7-1) may be supplemented for more understanding.

Section 3.2

(1) The XRD studies requires few clarifications: (i) is it possible to observe crystallized Bi₂O₃ if only Bi precursor are used in the similar experimental condition?; (ii) reflections of Bi₂O₃ and Bi₂WO₆ in the XRD patterns may be distinguished using symbols (like, asterisk '*' for Bi₂O₃); (iii) can pure Bi₂WO₆ formation is feasible if reaction time is prolonged for EA 1-1?.

(2) Typo: ration!

Section 3.6

(1) devolved!

Section 3.8

(1) The phrase 'excessive -NH₂ makes Bi³⁺ reduced to Bi⁰' may be supported with suitable literature.

(2) It should be 'adsorbed -OH groups' instead of 'absorbed'!

Conclusion

(1) The significant results obtained highlighting the role of ethanolamine may be presented.

Reviewed by: S. Girish Kumar, Asst. Professor

Dept. of Chemistry, CMR UNIVERSITY

KARNATAKA, INDIA.

<https://publons.com/author/1300703/s-girish-kumar#profile>

Review form: Reviewer 2

Is the manuscript scientifically sound in its present form?

Yes

Are the interpretations and conclusions justified by the results?

Yes

Is the language acceptable?

Yes

Is it clear how to access all supporting data?

Not Applicable

Do you have any ethical concerns with this paper?

No

Have you any concerns about statistical analyses in this paper?

No

Recommendation?

Major revision is needed (please make suggestions in comments)

Comments to the Author(s)

In this manuscript, the 3D flower-like Bi₂WO₆ was synthesized with the assist of ethanolamine through one-step microwave method and evaluated the photocatalytic property of samples. The results showed that the ethanolamine significantly influenced the morphology of Bi₂WO₆ products and improved the photocatalytic efficiency. Overall this topic is interesting, and the results were presented satisfactorily. However, some parts of the manuscript were not presented well, and revisions are needed prior to a possible publication in Royal Society Open Science. Detailed comments are listed as follows:

1. on Abstract: the authors should provide the detailed results and conclusions, and describe briefly if and how the structure and catalytic performance of Bi₂WO₆ change with the assist of ethanolamine.
2. On 1st paragraph: Better background information on microwave applications and organic pollutant removal by other methods including advanced oxidation and biological processes should be presented to the prospective audiences, and these following recent articles in this field could serve this purpose in some aspects: *Scientific Reports*, 2017, 7(1): 1668 (p1-8); *Chemical Engineering Journal*, 2018, 341: 126-136; *Chemical Engineering Journal*, 2018, 353: 533-541; *Chemosphere*, 2018, 200: 380-387; *Bioresource Technology*, 2018, 260: 196-203.
3. On 2nd paragraph: The use of photocatalysis for organic pollutants has received much interests. More recent articles on this topic should be reviewed, for example: *Microstructure and performance of Z-scheme photocatalyst of silver phosphate modified by MWCNTs and Cr-doped SrTiO₃ for malachite green degradation. Applied Catalysis B Environmental*, 2018, 227: 557-570.
4. On 3rd paragraph: Major properties of the model pollutant “methyl orange” should be mentioned. Why MO was selected as the model pollutant?
5. Line 48-53 in Page 3: The authors should consider to list some typical long chain compounds, compare them with ethanolamine, and describe the advantages and disadvantages.
6. In section 2.1, the symbol for temperature should be presented in an accepted style.
7. Line 30-41 in Page 6: Viscosity of EA is much higher than that of water, which could affect the mass transfer rate for ions in the system. This claim needs more reliable explanations and experimental evidences or data.
8. Line 25-27 in Page 7: More explanations and evidences including literature citation for “the high dispersion of EA makes the precursor sodium tungstate and bismuth nitrate to be fully in contact, thus contributing to the formation of Bi₂WO₆” are needed. The following literature could be referred for the dispersion: *Influences of anion concentration and valence on dispersion and aggregation of titanium dioxide nanoparticles in aqueous solutions. Journal of Environmental Sciences*, 2017, 54: 135-141.
9. Fig.2 shows that the EA 2-1 sample has the highest purity and crystallinity, but the authors only compared EA 7-1 to EA 5-1, indicated that the EA 7-1 sample has a higher purity and crystallinity. That comparison and explanation is incomplete and cannot support the following conclusion. In addition, the XRD spectra of ethanolamine should be provided to eliminate its effects for the crystal phase and structure of Bi₂WO₆.

10. Line 47-56 in Page 7: The authors should edit this manuscript more carefully. Such as Line 52-53 in Page 7, the clerical error in the sentence "This result proves that EA 2-1 is a mixture of two substances, which is consistent with the XRD result". it should be EA 1-1, not EA 2-1.

11. Line 55 in Page 8: The sentence "The surface area of all the samples is similar, shown in Table 1, which can indicate that the difference in adsorption capacity is not caused by the BET difference" should be placed in section 3.7 to explain the difference in adsorption capacity.

12. Line 3-19 in Page 11: the amount of EA could promote the increased atomic ratio of OS in Bi₂WO₆, the authors should explain it.

13. Line 55 in Page 9: in the paper, the values of band gaps of EA 0, EA 1-1, EA 2-1, EA 5-1 and EA 7-1 were calculated in the Fig.5, and the variation of energy band positions were observed from the XPS valence band spectra in the Fig.9. It's better to write out the calculation. In order to further confirm the energy band positions and its variation, other means, such as Mott-Schottky calculation process should be also provided. The literature mentioned in Comment 3 could provide such information.

14. On Conclusions: The results show that the addition of EA can improve the photocatalytic activity of materials and the increase of the amount of EA, the photocatalytic activity gradually increases. The author only compared the performances of EA 0, EA1-1, EA 2-1, EA 5-1 and EA 7-1, while did not conclude how the photocatalytic activity change when EA was increased further. Data on more higher proportion should be supplemented to find the optimal proportion, so as to make the conclusion and research more valuable.

Decision letter (RSOS-181422.R0)

13-Nov-2018

Dear Miss Liu:

Title: The 3-D flower-like Bi₂WO₆ assisted by ethanolamine through microwave method for efficiency photocatalytic activity

Manuscript ID: RSOS-181422

The editor assigned to your manuscript has now received comments from reviewers. We would like you to revise your paper in accordance with the referee and Subject Editor suggestions which can be found below (not including confidential reports to the Editor). Please note this decision does not guarantee eventual acceptance.

Please submit your revised paper before 06-Dec-2018. Please note that the revision deadline will expire at 00.00am on this date. If we do not hear from you within this time then it will be assumed that the paper has been withdrawn. In exceptional circumstances, extensions may be possible if agreed with the Editorial Office in advance. We do not allow multiple rounds of revision so we urge you to make every effort to fully address all of the comments at this stage. If deemed necessary by the Editors, your manuscript will be sent back to one or more of the original reviewers for assessment. If the original reviewers are not available we may invite new reviewers.

Please also include the following statements alongside the other end statements. As we cannot publish your manuscript without these end statements included, if you feel that a given heading is not relevant to your paper, please nevertheless include the heading and explicitly state that it is not relevant to your work.

- Ethics statement

Please clarify whether you received ethical approval from a local ethics committee to carry out your study. If so please include details of this, including the name of the committee that gave consent in a Research Ethics section after your main text. Please also clarify whether you received informed consent for the participants to participate in the study and state this in your Research Ethics section.

OR

Please clarify whether you obtained the necessary licences and approvals from your institutional animal ethics committee before conducting your research. Please provide details of these licences and approvals in an Animal Ethics section after your main text.

OR

Please clarify whether you obtained the appropriate permissions and licences to conduct the fieldwork detailed in your study. Please provide details of these in your methods section.

On behalf of the Subject Editor Professor Anthony Stace and the Associate Editor Professor Eva Hevia.

RSC Associate Editor:
Comments to the Author:

(There are no comments.)

RSC Subject Editor:

Comments to the Author:

(There are no comments.)

Reviewers' Comments to Author:

Reviewer: 1

Comments to the Author(s)

Dear authors,

The specific insights into the exceptional role of ethanolamine on the morphological evolution of Bi₂WO₆ sheets and further assembling into flower-like structure are explained in detail. The further discussions related to the shifting of band edge positions, photoluminescence spectroscopy and Mott-Schottky plots that substantiates the influence of ethanolamine are the other interesting aspects. However, few clarifications and thorough language editing are still essential for improved understanding.

Major comments

(1) The recycling ability of the photocatalyst may be discussed.

Minor comments

Title: It may be modified for reader's attention (up to the author's choice): (i) Pivotal role of ethanolamine on the structure-electronic and photocatalytic properties of flower-like Bi₂WO₆; (ii) Role of ethanolamine on the morphological evolution of 3-D flower like Bi₂WO₆ photocatalyst by microwave method; (iii) Formation mechanism of ethanolamine assisted 3-D flower like Bi₂WO₆ photocatalyst under microwave irradiation; (iv) 3-D flower like Bi₂WO₆: Insight into the role of ethanolamine on the formation mechanisms and photocatalytic property.

Abstract

(1) Indicate the reaction temperature and time within the parenthesis after the sentence (microwave method) and specify concentration of ethanolamine for clarity.

(2) The role of ethanolamine in altering the pressure of the reaction and on the crystallization of Bi₂WO₆ must be highlighted.

(3) The results derived from materials characterization should be briefly emphasized.

Section 1.0

(1) Modify the starting phrase 'In recently, pollution!' (avoid using pollution twice in the same phrase).

(1) Add few review articles related to Bi₂WO₆ along with ref. [2,3]: Applied Surface Science 355 (2015) 939-958; Catalysis Science and Technology 2 (2012) 694-706; Chemical Society Reviews 43 (2014) 5276-5287.

(2) Remove 'more than 10 hours' and generalize as 'longer time', while discussing hydrothermal method.

Section 2.1

(1) The preparative method should be detailed out completely to ensure reproducibility: (i) volume of Bi precursors and tungstate precursors taken should be indicated; (ii) concentration of ethanolamine must be specified.

(2) The apparatus in which these solutions were mixed must be indicated.

(3) The samples may be abbreviated as 'EA:X' instead of 'EA X'.

Section 2.3

- (1) The details of excitation source including its intensity, wavelength maxima and distance between light source and solution surface must be provided.
- (2) Specify the solution pH at which degradation reaction were carried out.

Section 3.1

- (1) The effect on pressure on the reaction speed as referenced from literature [15,16] may be briefly explained.
- (2) The viscosity of the ethanolamine-water mixture (EA 1-1 to EA 7-1) may be supplemented for more understanding.

Section 3.2

- (1) The XRD studies requires few clarifications: (i) is it possible to observe crystallized Bi_2O_3 if only Bi precursor are used in the similar experimental condition?; (ii) reflections of Bi_2O_3 and Bi_2WO_6 in the XRD patterns may be distinguished using symbols (like, asterisk '*' for Bi_2O_3); (iii) can pure Bi_2WO_6 formation is feasible if reaction time is prolonged for EA 1-1?.
- (2) Typo: ration!

Section 3.6

- (1) devolved!

Section 3.8

- (1) The phrase 'excessive $-\text{NH}_2$ makes Bi^{3+} reduced to Bi^0 ' may be supported with suitable literature.
- (2) It should be 'adsorbed $-\text{OH}$ groups' instead of 'absorbed'!

Conclusion

- (1) The significant results obtained highlighting the role of ethanolamine may be presented.

Reviewed by: S. Girish Kumar, Asst. Professor

Dept. of Chemistry, CMR UNIVERSITY

KARNATAKA, INDIA.

<https://publons.com/author/1300703/s-girish-kumar#profile>

Reviewer: 2

Comments to the Author(s)

In this manuscript, the 3D flower-like Bi_2WO_6 was synthesized with the assist of ethanolamine through one-step microwave method and evaluated the photocatalytic property of samples. The results showed that the ethanolamine significantly influenced the morphology of Bi_2WO_6 products and improved the photocatalytic efficiency. Overall this topic is interesting, and the results were presented satisfactorily. However, some parts of the manuscript were not presented well, and revisions are needed prior to a possible publication in Royal Society Open Science. Detailed comments are listed as follows:

1. on Abstract: the authors should provide the detailed results and conclusions, and describe briefly if and how the structure and catalytic performance of Bi_2WO_6 change with the assist of ethanolamine.
2. On 1st paragraph: Better background information on microwave applications and organic pollutant removal by other methods including advanced oxidation and biological processes

should be presented to the prospective audiences, and these following recent articles in this field could serve this purpose in some aspects: *Scientific Reports*, 2017, 7(1): 1668 (p1-8); *Chemical Engineering Journal*, 2018, 341: 126-136; *Chemical Engineering Journal*, 2018, 353: 533-541; *Chemosphere*, 2018, 200: 380-387; *Bioresource Technology*, 2018, 260: 196-203.

3. On 2nd paragraph: The use of photocatalysis for organic pollutants has received much interests. More recent articles on this topic should be reviewed, for example: Microstructure and performance of Z-scheme photocatalyst of silver phosphate modified by MWCNTs and Cr-doped SrTiO₃ for malachite green degradation. *Applied Catalysis B Environmental*, 2018, 227: 557-570.

4. On 3rd paragraph: Major properties of the model pollutant "methyl orange" should be mentioned. Why MO was selected as the model pollutant?

5. Line 48-53 in Page 3: The authors should consider to list some typical long chain compounds, compare them with ethanolamine, and describe the advantages and disadvantages.

6. In section 2.1, the symbol for temperature should be presented in an accepted style.

7. Line 30-41 in Page 6: Viscosity of EA is much higher than that of water, which could affect the mass transfer rate for ions in the system. This claim needs more reliable explanations and experimental evidences or data.

8. Line 25-27 in Page 7: More explanations and evidences including literature citation for "the high dispersion of EA makes the precursor sodium tungstate and bismuth nitrate to be fully in contact, thus contributing to the formation of Bi₂WO₆" are needed. The following literature could be referred for the dispersion: Influences of anion concentration and valence on dispersion and aggregation of titanium dioxide nanoparticles in aqueous solutions. *Journal of Environmental Sciences*, 2017, 54: 135-141.

9. Fig.2 shows that the EA 2-1 sample has the highest purity and crystallinity, but the authors only compared EA 7-1 to EA 5-1, indicated that the EA 7-1 sample has a higher purity and crystallinity. That comparison and explanation is incomplete and cannot support the following conclusion. In addition, the XRD spectra of ethanolamine should be provided to eliminate its effects for the crystal phase and structure of Bi₂WO₆.

10. Line 47-56 in Page 7: The authors should edit this manuscript more carefully. Such as Line 52-53 in Page 7, the clerical error in the sentence "This result proves that EA 2-1 is a mixture of two substances, which is consistent with the XRD result". it should be EA 1-1, not EA 2-1.

11. Line 55 in Page 8: The sentence "The surface area of all the samples is similar, shown in Table 1, which can indicate that the difference in adsorption capacity is not caused by the BET difference" should be placed in section 3.7 to explain the difference in adsorption capacity.

12. Line 3-19 in Page 11: the amount of EA could promote the increased atomic ratio of OS in Bi₂WO₆, the authors should explain it.

13. Line 55 in Page 9: in the paper, the values of band gaps of EA 0, EA 1-1, EA 2-1, EA 5-1 and EA 7-1 were calculated in the Fig.5, and the variation of energy band positions were observed from the XPS valence band spectra in the Fig.9. It's better to write out the calculation. In order to further confirm the energy band positions and its variation, other means, such as Mott-Schottky calculation process should be also provided. The literature mentioned in Comment 3 could provide such information.

14. On Conclusions: The results show that the addition of EA can improve the photocatalytic activity of materials and the increase of the amount of EA, the photocatalytic activity gradually increases. The author only compared the performances of EA 0, EA1-1, EA 2-1, EA 5-1 and EA 7-1, while did not conclude how the photocatalytic activity change when EA was increased further. Data on more higher proportion should be supplemented to find the optimal proportion, so as to make the conclusion and research more valuable.

Author's Response to Decision Letter for (RSOS-181422.R0)

See Appendix A.

RSOS-181422.R1 (Revision)

Review form: Reviewer 1 (Girish Kumar)

Is the manuscript scientifically sound in its present form?

Yes

Are the interpretations and conclusions justified by the results?

Yes

Is the language acceptable?

No

Is it clear how to access all supporting data?

Not Applicable

Do you have any ethical concerns with this paper?

No

Have you any concerns about statistical analyses in this paper?

I do not feel qualified to assess the statistics

Recommendation?

Accept with minor revision (please list in comments)

Comments to the Author(s)

Dear authors,

The point-to-point response for all the comments deserves appreciation and revised manuscript provides more insight into the work. The listed comments may be considered for better readability.

Minor comments

Title: Replace 'efficiency' by 'efficient'!

Abstract

(1) Remove the phrase 'The whole process took only ten minutes'!

Introduction

(1) The ethanolamine should be abbreviated as 'EA' in its first appearance in the text.

Section 2.3

(1) Is it 460 or 464 nm! It may be verified.

Section 3.0

(1) It should be 'Figure' instead of 'figure' throughout.

(2) The term 'compound rates' could not be followed. (near ref. 30 and at other places as well)

(3) The spin orbit coupling values must be made subscript to the orbital in the XPS studies.

(4) Figure 7-2?

(5) It should be 'XRD patterns' instead of 'XRD images'!

Reviewed by: S. Girish Kumar, Asst. Professor

Dept. of Chemistry, CMR UNIVERSITY

KARNATAKA, INDIA.

<https://publons.com/author/1300703/s-girish-kumar#profile>

Review form: Reviewer 2

Is the manuscript scientifically sound in its present form?

Yes

Are the interpretations and conclusions justified by the results?

Yes

Is the language acceptable?

Yes

Is it clear how to access all supporting data?

Yes

Do you have any ethical concerns with this paper?

No

Have you any concerns about statistical analyses in this paper?

No

Recommendation?

Accept with minor revision (please list in comments)

Comments to the Author(s)

The authors have either revised this submission according to the comments and suggestions from the reviewers satisfactorily, or responded to the concerns from the reviewers well. I suggest this

manuscript be accepted for publication in Royal Society Open Science after the authors correct the following problems.

1. There are many spelling or style problems in the reference list. For example, some journals were presented with whole names (such as References 1 and 2), and some with abbreviated ones (such as References 3 and 4).
2. Some of whole journal titles were listed in upper case (such as References 3 and 4), and some not (such as References 5 and 6).
3. Some bibliographic information are lost, such as Reference 12 whose volume and page numbers of (Chem Eng J, 2018, 353: 533-541) were not provided.

The authors should correct these minor mistakes to meet the standard of publication.

Decision letter (RSOS-181422.R1)

14-Jan-2019

Dear Miss Liu:

Title: The 3-D flower-like Bi₂WO₆ assisted by ethanolamine through microwave method for efficiency photocatalytic activity
Manuscript ID: RSOS-181422.R1

Thank you for submitting the above manuscript to Royal Society Open Science. On behalf of the Editors and the Royal Society of Chemistry, I am pleased to inform you that your manuscript will be accepted for publication in Royal Society Open Science subject to minor revision in accordance with the referee suggestions. Please find the reviewers' comments at the end of this email.

The reviewers and handling editors have recommended publication, but also suggest some minor revisions to your manuscript. Therefore, I invite you to respond to the comments and revise your manuscript.

Because the schedule for publication is very tight, it is a condition of publication that you submit the revised version of your manuscript before 23-Jan-2019. Please note that the revision deadline will expire at 00.00am on this date. If you do not think you will be able to meet this date please let me know immediately.

- 1) A text file of the manuscript (tex, txt, rtf, docx or doc), references, tables (including captions) and figure captions. Do not upload a PDF as your "Main Document".

- 2) A separate electronic file of each figure (EPS or print-quality PDF preferred (either format should be produced directly from original creation package), or original software format)
- 3) Included a 100 word media summary of your paper when requested at submission. Please ensure you have entered correct contact details (email, institution and telephone) in your user account
- 4) Included the raw data to support the claims made in your paper. You can either include your data as electronic supplementary material or upload to a repository and include the relevant doi within your manuscript
- 5) All supplementary materials accompanying an accepted article will be treated as in their final form. Note that the Royal Society will neither edit nor typeset supplementary material and it will be hosted as provided. Please ensure that the supplementary material includes the paper details where possible (authors, article title, journal name).

Best wishes,

Dr Laura Smith
Publishing Editor, Journals

On behalf of the Subject Editor Professor Anthony Stace and the Associate Editor Professor Eva Hevia.

RSC Associate Editor:
Comments to the Author:
(There are no comments.)

RSC Subject Editor:
Comments to the Author:
(There are no comments.)

Reviewer comments to Author:

Reviewer: 1

Comments to the Author(s)

Dear authors,

The point-to-point response for all the comments deserves appreciation and revised manuscript provides more insight into the work. The listed comments may be considered for better readability.

Minor comments

Title: Replace 'efficiency' by 'efficient'!

Abstract

(1) Remove the phrase 'The whole process took only ten minutes'!

Introduction

(1) The ethanolamine should be abbreviated as 'EA' in its first appearance in the text.

Section 2.3

(1) Is it 460 or 464 nm! It may be verified.

Section 3.0

(1) It should be 'Figure' instead of 'figure' throughout.

(2) The term 'compound rates' could not be followed. (near ref. 30 and at other places as well)

(3) The spin orbit coupling values must be made subscript to the orbital in the XPS studies.

(4) Figure 7-2?

(5) It should be 'XRD patterns' instead of 'XRD images'!

Reviewed by: S. Girish Kumar, Asst. Professor

Dept. of Chemistry, CMR UNIVERSITY

KARNATAKA, INDIA.

<https://publons.com/author/1300703/s-girish-kumar#profile>

Reviewer: 2

Comments to the Author(s)

The authors have either revised this submission according to the comments and suggestions from the reviewers satisfactorily, or responded to the concerns from the reviewers well. I suggest this manuscript be accepted for publication in Royal Society Open Science after the authors correct the following problems.

1. There are many spelling or style problems in the reference list. For example, some journals were presented with whole names (such as References 1 and 2), and some with abbreviated ones (such as References 3 and 4).

2. Some of whole journal titles were listed in upper case (such as References 3 and 4), and some not (such as References 5 and 6).

3. Some bibliographic information are lost, such as Reference 12 whose volume and page numbers of (Chem Eng J, 2018, 353: 533-541) were not provided.

The authors should correct these minor mistakes to meet the standard of publication.

Author's Response to Decision Letter for (RSOS-181422.R1)

See Appendix B.

Decision letter (RSOS-181422.R2)

21-Jan-2019

Dear Miss Liu:

Title: The 3-D flower-like Bi₂WO₆ assisted by ethanalamine through microwave method for efficiency photocatalytic activity
Manuscript ID: RSOS-181422.R2

It is a pleasure to accept your manuscript in its current form for publication in Royal Society Open Science. The chemistry content of Royal Society Open Science is published in collaboration with the Royal Society of Chemistry.

On behalf of the Subject Editor Professor Anthony Stace and the Associate Editor Professor Eva Hevia.

RSC Associate Editor
Comments to the Author:
(There are no comments.)

Reviewer(s)' Comments to Author:

Appendix A

Dear Editor and Reviewers:

Thank you for your letter and for the reviewers' comments concerning our manuscript entitled "The 3-D flower-like Bi₂WO₆ assisted by ethanolamine through microwave method for efficiency photocatalytic activity". Those comments are all valuable and very helpful for revising and improving our paper, as well as the important guiding significance to our researches. We have studied comments carefully and have made correction which we hope meet with approval. Revised portion are marked in red in the paper. The main corrections in the paper and the responds to the reviewer's comments are as following:

Responds to the reviewer's comments:

Reviewer #1:

1. Response to comment: Abstract(1) Indicate the reaction temperature and time within the parenthesis after the sentence (microwave method) and specify concentration of ethanolamine for clarity. (2) The role of ethanolamine in altering the pressure of the reaction and on the crystallization of Bi₂WO₆ must be highlighted. (3) The results derived from materials characterization should be briefly emphasized.

Response: We have made corrections according to the Reviewer's comments. All the corrections were written in red.

2. Response to comment: Section 1.0(1) Modify the starting phrase 'In recently, pollution!' (avoid using pollution twice in the same phrase). (1) Add few review articles related to Bi₂WO₆ along with ref. [2,3]: Applied Surface Science 355 (2015) 939-958; Catalysis Science and Technology 2 (2012) 694-706; Chemical Society Reviews 43 (2014) 5276-5287.(2) Remove 'more than 10 hours' and generalize as 'longer time', while discussing hydrothermal method.

Response: We have made corrections according to the Reviewer's comments and mentioned references have been added. All the corrections were written in red.

3. Response to comment: Section 2.1(1) The preparative method should be detailed out completely to ensure reproducibility: (i) volume of Bi precursors and tungstate precursors taken should be indicated; (ii) concentration of ethanolamine must be specified. (2) The apparatus in which these solutions were mixed must be indicated. (3) The samples may be abbreviated as 'EA:X' instead of 'EA X'.

Response: Thank you for your comment. (1) (i) The amount of Bi precursors and tungstate precursors has been shown in the paper, which is 2 mmol and 1 mmol, respectively. (ii) the concentration of used ethanolamine is about 99%, and which has been indicated in the article. (2) The whole experiment was going on microwave reactor (Biotage Sweden). No additional mixing apparatus were used. (3) We have re-written this part according to the Reviewer's suggestion. All the corrections were written in red.

4. Response to comment: Section 2.3(1) The details of excitation source including its intensity, wavelength maxima and distance between light source and solution surface must be provided. (2) Specify the solution pH at which degradation reaction were carried out.

Response: The details of excitation source and the solution pH (pH=7) have been added in the article, which were written in red.

5. Section 3.2(1) The XRD studies requires few clarifications: (i) is it possible to observe crystallized Bi₂O₃ if only Bi precursor are used in the similar experimental condition?; (ii) reflections of Bi₂O₃ and Bi₂WO₆ in the XRD patterns may be distinguished using symbols (like, asterisk '*' for Bi₂O₃); (iii) can pure Bi₂WO₆ formation is feasible if reaction time is prolonged for EA 1-1?.

(2) Typo: ration!

Response: (1) Thank you for your comment. (i) We have supplemented this data. The crystallized Bi₂O₃ was obtained if only Bi precursor is used in the similar experimental condition (Fig.S1). (ii) We have re-written this part according to the Reviewer's suggestion. All the corrections were written in red. (iii) We have supplemented this data. When the reaction time is prolonged for EA

1-1, the diffraction peaks characteristic of Bi₂O₃ gradually decreased, and that of Bi₂WO₆ enhanced gradually (Fig.S2)

(2) We have re-written this part according to the Reviewer's suggestion. All the corrections were written in red.

6. Section 3.6 (1) devolved!

Response: We have re-written this part according to the Reviewer's suggestion. All the corrections were written in red.

7. Section 3.8(1) The phrase 'excessive -NH₂ makes Bi³⁺ reduced to Bi⁰' may be supported with suitable literature. (2) It should be 'adsorbed -OH groups' instead of 'absorbed'!

Response:

(1) Thank you for your comment. Since -NH₂ is reducible, excessive -NH₂ makes Bi³⁺ reduced to Bi⁰, Our experiments prove this point (Fig. 11). In addition, we also added relevant references.

(2) We have re-written this part according to the Reviewer's suggestion. All the corrections were written in red.

8. Conclusion(1) The significant results obtained highlighting the role of ethanolamine may be presented.

Response: The role of ethanolamine has been added in the article, which was written in red.

Reviewer #2:

1. on Abstract: the authors should provide the detailed results and conclusions, and describe briefly if and how the structure and catalytic performance of Bi₂WO₆ change with the assist of ethanolamine.

Response: We have re-written this part according to the Reviewer's suggestion. All the corrections were written in red.

2. On 1st paragraph: Better background information on microwave applications and organic pollutant removal by other methods including advanced oxidation and biological processes should be presented to the prospective audiences, and these following recent articles in this field could serve this purpose in some aspects: Scientific Reports, 2017, 7(1): 1668 (p1-8); Chemical

Engineering Journal, 2018, 341: 126-136; Chemical Engineering Journal, 2018, 353: 533-541; Chemosphere, 2018, 200: 380-387; Bioresource Technology, 2018, 260: 196-203.

Response: We have made corrections according to the Reviewer's comments and mentioned references have been added. All the corrections were written in red.

3. On 2nd paragraph: The use of photocatalysis for organic pollutants has received much interests. More recent articles on this topic should be reviewed, for example: Microstructure and performance of Z-scheme photocatalyst of silver phosphate modified by MWCNTs and Cr-doped SrTiO₃ for malachite green degradation. Applied Catalysis B Environmental, 2018, 227: 557-570.

Response: We have made corrections according to the Reviewer's comments and mentioned references have been added. All the corrections were written in red.

4. On 3rd paragraph: Major properties of the model pollutant "methyl orange" should be mentioned. Why MO was selected as the model pollutant?

Response: Thank you for your comment. Methyl orange (MO) is stable and does not decompose under the action of light. Moreover, the degradation rate of MO can be calculated by absorbance and concentration standard curves. Therefore, MO is used as a model for photocatalytic experiments. We have explained this point in the article. All the corrections were written in red.

5. Line 48-53 in Page 3: The authors should consider to list some typical long chain compounds, compare them with ethanolamine, and describe the advantages and disadvantages.

Response: We have re-written this part according to the Reviewer's suggestion. All the corrections were written in red.

6. In section 2.1, the symbol for temperature should be presented in an accepted style.

Response: We have made corrections according to the Reviewer's comments. All the corrections were written in red.

7. Line 30-41 in Page 6: Viscosity of EA is much higher than that of water, which could affect the mass transfer rate for ions in the system. This claim needs more reliable explanations and experimental evidences or data.

Response: Thank you for your comment. We have added the relevant references have been added (25). All the corrections were written in red.

8. Line 25-27 in Page 7: More explanations and evidences including literature citation for “the high dispersion of EA makes the precursor sodium tungstate and bismuth nitrate to be fully in contact, thus contributing to the formation of Bi₂WO₆” are needed. The following literature could be referred for the dispersion: Influences of anion concentration and valence on dispersion and aggregation of titanium dioxide nanoparticles in aqueous solutions. *Journal of Environmental Sciences*, 2017, 54: 135-141.

Response: We have made corrections according to the Reviewer’s comments and mentioned references have been added. All the corrections were written in red.

9. Fig.2 shows that the EA 2-1 sample has the highest purity and crystallinity, but the authors only compared EA 7-1 to EA 5-1, indicated that the EA 7-1 sample has a higher purity and crystallinity. That comparison and explanation is incomplete and cannot support the following conclusion. In addition, the XRD spectra of ethanolamine should be provided to eliminate its effects for the crystal phase and structure of Bi₂WO₆.

Response: Thank you for your comment. In this paper, we focus on the 3D flower-like structure. Seen from Fig.3, since only EA 5-1 and EA 7-1 samples formed the 3D flower-like structure, we only compared EA 7-1 to EA 5-1. We have omitted the expression in this part in the article. In addition, since EA is liquid at room temperature, the XRD spectrum of which has no peak, there is no interference with the crystal phase and structure of Bi₂WO₆. So we think it is not necessary to supplement its XRD spectrum

10. Line 47-56 in Page 7: The authors should edit this manuscript more carefully. Such as Line 52-53 in Page 7, the clerical error in the sentence “This result proves that EA 2-1 is a mixture of two

substances, which is consistent with the XRD result". it should be EA 1-1, not EA 2-1.

Response: Thank you for your comment. We have made corrections according to the Reviewer's comments and mentioned references have been added. All the corrections were written in red.

11. Line 55 in Page 8: The sentence "The surface area of all the samples is similar, shown in Table 1, which can indicate that the difference in adsorption capacity is not caused by the BET difference" should be placed in section 3.7 to explain the difference in adsorption capacity.

Response: Thank you for your comment. We have made corrections according to the Reviewer's comments and mentioned references have been added. All the corrections were written in red.

12. Line 3-19 in Page 11: the amount of EA could promote the increased atomic ratio of OS in Bi₂WO₆, the authors should explain it.

Response: Thank you for your comment. The O 1s core level at 532.25 eV (OS) can be ascribed to the oxygen in -OH. EA molecules contain active hydroxyl and secondary amino group. With the proportion of EA increasing through hydrogen bonding, the association of -OH and Bi₂WO₆ molecular becomes greater. So we think the amount of EA could promote the increased atomic ratio of OS in Bi₂WO₆. We have explained this reason in article 3.8.

13. Line 55 in Page 9: in the paper, the values of band gaps of EA 0, EA 1-1, EA 2-1, EA 5-1 and EA 7-1 were calculated in the Fig.5, and the variation of energy band positions were observed from the XPS valence band spectra in the Fig.9. It's better to write out the calculation. In order to further confirm the energy band positions and its variation, other means, such as Mott-Schottky calculation process should be also provided. The literature mentioned in Comment 3 could provide such information.

Response: Thank you for your comment. The XPS valence band spectra of EA 7-1, EA 5-1 and EA 2-1, the VB potentials are 1.77 eV, 1.81 eV and 1.97 eV, respectively. In addition, as shown in Fig. 10, the flat band potentials are estimated to be -0.67 V, -0.65 V, -0.59 V for

EA 2-1, EA 5-1 and EA 7-1 respectively. We have already explained this in article.

14. On Conclusions: The results show that the addition of EA can improve the photocatalytic activity of materials and the increase of the amount of EA, the photocatalytic activity gradually increases. The author only compared the performances of EA 0, EA1-1, EA 2-1, EA 5-1 and EA 7-1, while did not conclude how the photocatalytic activity change when EA was increased further. Data on more higher proportion should be supplemented to find the optimal proportion, so as to make the conclusion and research more valuable.

Response: Thank you for your comment. We have supplemented this data, as shown in Fig.S4.

Appendix B

Dear Editor and Reviewers:

Thank you for your letter and for the reviewers' comments concerning our manuscript entitled "The 3-D flower-like Bi₂WO₆ assisted by ethanolamine through microwave method for efficiency photocatalytic activity". Those comments are all valuable and very helpful for revising and improving our paper, as well as the important guiding significance to our researches. We have studied comments carefully and have made correction which we hope meet with approval. Revised portion are marked in red in the paper. The main corrections in the paper and the responds to the reviewer's comments are as flowing:

Responds to the reviewer's comments:

Reviewer #1:

1. Response to comment: Title: Replace 'efficiency' by 'efficient'!

Response: We have made corrections according to the Reviewer's comments. All the corrections were written in red.

2. Response to comment: Abstract(1) Remove the phrase 'The whole process took only ten minutes'!

Response: We have already removed the phrase in abstract.

3. Response to comment: Introduction (1) The ethanolamine should be abbreviated as 'EA' in its first appearance in the text.

Response: We have already made corrections according to the Reviewer's comments. All the corrections were written in red.

4. Response to comment: Section 2.3 (1) Is it 460 or 464 nm! It may be verified.

Response: We have already changed the maximum absorption wavelength as 464 nm.

5. Response to comment: Section 3.0

(1) It should be 'Figure' instead of 'figure' throughout.

(2) The term 'compound rates' could not be followed. (near ref. 30 and at other places as well)

(3) The spin orbit coupling values must be made subscript to the orbital in the XPS studies.

(4) Figure 7-2?

(5) It should be 'XRD patterns' instead of 'XRD images'!

Response: (1) We have already made corrections in 3.2. The correction was written in red.

(2) We have already changed "compound rates" to "recombination rate". The correction was written in red.

(3) We have already labeled the spin orbital coupling value in Fig.6 and the corresponding explanations have been in 3.6.

(4) We have already changed "Figure 7-2" to "Figure 7 (b)". The correction was written in red.

(5) We have made corrections according to the Reviewer's comments. The correction was written in red.

Reviewer #2:

Response to comment:

1. There are many spelling or style problems in the reference list. For example, some journals were presented with whole names (such as References 1 and 2), and some with abbreviated ones (such as References 3 and 4).

2. Some of whole journal titles were listed in upper case (such as References 3 and 4), and some not (such as References 5 and 6).

3. Some bibliographic information are lost, such as Reference 12 whose volume and page numbers of (Chem Eng J, 2018, 353: 533-541) were not provided.

The authors should correct these minor mistakes to meet the standard of publication.

Response: We have made corrections according to the Reviewer's comments in the reference list. The correction was written in red.